# Retrograde Cricopharyngeal Dysfunction: An Update of Pathophysiological Mechanisms and Future Directions

**DOI:** 10.3390/toxins18010008

**Published:** 2025-12-22

**Authors:** Marie Mailly, Jerome R. Lechien

**Affiliations:** 1Department of Otolaryngology-Head and Neck Surgery, Foch Hospital, Paris Saclay University, 91190 Paris, France; marie.mailly@gmail.com; 2Department of Surgery, UMONS Research Institute for Language Science and Technology, University of Mons (UMons), 7000 Mons, Belgium

**Keywords:** retrograde, cricopharyngeal, cricopharyngeus, dysfunction, otolaryngology, botulinum toxin

## Abstract

This scoping review aimed to summarize the current literature on the etiological and pathophysiological mechanisms associated with the development of retrograde cricopharyngeus dysfunction (R-CPD) through a PRISMA literature search. According to the current literature, a family history of R-CPD was reported in 28.0% of patients across studies, with childhood onset in 55.5% of cases. Gastroesophageal reflux disease and laryngopharyngeal reflux disease prevalence in R-CPD patients ranged from 16.3 to 51.9%, with improvement of heartburn symptoms after treatment. High-resolution manometry revealed dysmotility disorders in 43.5–80.0% of patients, with absent peristalsis in 11–25%. Carbonated drink provocative testing provided diagnostic usefulness in patients with unclear diagnoses by demonstrating failure of cricopharyngeal sphincter relaxation for retrograde gas. Notably, 75.5–79.9% of patients maintained symptom relief beyond the expected pharmacologic duration of botulinum toxin (approximately 6 months), suggesting potential neuroplastic adaptation or learned compensatory mechanisms in overcoming retrograde cricopharyngeal sphincter dysfunction. The pathophysiology of R-CPD remains incompletely understood, with a lack of epidemiological and pediatric studies. The genetic and environmental factors may play a key role, but future studies are needed to clarify their roles in the development of R-CPD.

## 1. Introduction

The first cases of retrograde cricopharyngeal dysfunction (R-CPD) were published at the end of the 1980s [1,2], while the first case series identifying R-CPD as a specific syndrome was just published in 2019 [3]. In this first case series of patients with an inability to burp, Bastian and Smithson proposed that R-CPD is related to the loss of the belch reflex, which leads to troublesome symptoms [3]. This reflex is normally a physiological response to gaseous pressurization of the esophagus. Such pressurization occurs when transient lower esophageal sphincter relaxation allows gas from the stomach to reach the esophagus. If the upper esophageal sphincter fails to relax during this pressurization, gas becomes trapped in the esophagus. This trapped gas is subsequently carried back into the stomach by peristalsis in individuals without significant dysmotility, while in individuals with lost esophageal peristalsis, the gas can be blocked in the esophagus. Both retrograde processes may cause characteristic symptoms of varying severity. These symptoms include gurgling noises, bloating, chest pain, and excessive flatulence, which are typically accompanied by the inability to burp [3,4,5]. The fiberoptic examination of R-CPD patients is commonly unremarkable; however, some patients report a “tunnel sign” at swallowing during the nasofibroscopy (Figure 1). The tunnel sign reflects entrapped gas in the esophagus that cannot be eructated. To date, the etiology of R-CPD remains unelucidated, with recent studies suggesting a potential role of childhood reflux disease in cricopharyngeal and upper aerodigestive tract irritation and related protective airway mechanisms (cricopharyngeal muscle hypercontractility) [4,5].

The aim of this scoping review is to summarize the current literature on the etiological and pathophysiological mechanisms associated with the development of R-CPD.

## 2. Methods

The criteria for study inclusion and exclusion were based on the population, intervention, comparison, outcome, timing, and setting (PICOTS) framework [6]. The data review and collection were performed by two independent authors (J.R.L. and M.M.) following the PRISMA checklist for scoping reviews (Appendix A) [7].

**Population, inclusion, and exclusion criteria**: Studies had to report medical record information (comorbidities) or functional evaluations (e.g., high-resolution manometry, impedance-pH monitoring, and electromyography) of patients with a clear diagnosis of R-CPD. R-CPD diagnosis was based on the inability to belch associated with troublesome symptoms, such as chest pain/bloating, gurgling noises, hiccups, and excessive flatulence [3,4]. Because some patients have an incomplete clinical picture [4], the inability to belch was considered necessary for the diagnosis, whereas the other symptoms supported the diagnosis.

**Outcomes**: The following primary outcomes were considered: time of R-CPD occurrence (childbirth vs. throughout life), comorbidities, and functional assessments of esophagus or cricopharyngeal sphincter (e.g., high-resolution manometry, impedance-pH monitoring, electromyography, barium swallow, and endoscopy). The secondary outcomes included study design, number of patients, gender ratio, age (mean/median), diagnosis method, medical or surgical procedure(s), and posttreatment outcomes.

**Intervention and comparison**: If the pre- to posttreatment outcomes could provide relevant information for the pathophysiology underlying the development of R-CPD, the authors extracted the following procedure outcomes from selected studies: approach to botulinum toxin injection into the cricopharyngeus muscle (transnasal, EMG-guided transcervical, or in operating room), sphincter balloon dilatation, or surgical myotomy.

**Timing and Setting**: There were no criteria for specific stages or timing in the “disease process” of the study population.

**Search strategy**: The literature research was conducted through PubMed, Scopus, and Cochrane Library databases for relevant peer-reviewed publications related to etiology and pathophysiological findings of R-CPD. The MeSH and non-MeSH keywords “Cricopharyngeus”, “Sphincter”, “Belching”, “Treatment”, “Management” and “Cricopharyngeal”, “Dysfunction”, “Disorder”, “Belch”, “Inability”, and “Procedure” were used to identify case reports, clinical prospective/retrospective studies, reviews, and meta-analyses. Case reports were excluded. The overlap between clinical studies published by the same teams was evaluated, and the smallest cohorts were excluded for the overlapped outcomes. The papers had full texts or titles containing the search terms. The findings were reviewed for relevance, and potential reference lists of these articles were examined for additional pertinent studies. Ethics committee approval was not required.

## 3. Discussion

### 3.1. Genetic Pattern

To date, there are no genetic analysis studies conducted on R-CPD. The potential influence of genetics was indirectly investigated through the proportion of patients with a childhood history of R-CPD, the existence of family history, and ethnic patterns. A family history was prospectively or systematically investigated in four studies [5,8,9,10]. The patients recruited in the studies of Hoesli et al. and Mailly et al. systematically completed a questionnaire to identify potential family and childhood histories of R-CPD [5,8]. In a cohort of 200 patients, Hoesli et al. identified that 84 (41.8%) patients had family members with R-CPD [8], while the family proportion of R-CPD was 17% (*n* = 18/106) in the French study [4]. The difference across the two studies was much higher for childhood R-CPD history because 196 (98%) vs. 28 (26.4%) patients reported having childhood R-CPD in the American and European studies, respectively [5,8]. In the case series of Miller et al., there was no patient with a family history, but all (*n* = 13) had a childhood history [10]. The family history was investigated in a cross-sectional study conducted on social media platforms by Chen et al., who reported 21.6% (*n* = 43/199) prevalence of family history among an international heterogeneous sample of R-CPD patients [9]. Considering studies reporting family history, R-CPD was found in at least one other family member in 28.0% of cases (Table 1).

The time of onset of R-CPD was reported in eight papers (Table 1) [5,8,9,10,11,12,13,14]. The mean proportion of childhood R-CPD history was 55.5%, ranging from 22.7% to 100%. Among patients without childhood R-CPD history, the symptom onset was primarily in adolescence and, rarely, in adulthood (Table 1). The exact age of first consultation was around 30 years, with many patients being diagnosed in their 20s and 30s. The relationship between R-CPD and ethnicity was indirectly explored in three studies [9,15,16]. Chen et al. reported a majority of White (88.5%) and Asian (6.8%) patients, with only 0.5% of Black patients responding to the international R-CPD survey [9]. In the prospective cohort study of Yousef et al., 85% of patients were White, whereas there were no Hispanic or Black people [15]. Doruk et al. treated 52/57 (91.2%), 4/57 (7%), and 1/57 (1.8%) White, Asian, and Black patients, respectively [16]. A recent systematic review suggested that the White, Black, and Asian proportions of R-CPD patients ranged from 85.0% to 91.3%, 0.0% to 1.7%, and 0.0% to 7.0%, respectively [4]. Importantly, except for the survey by Chen et al. [9], these findings may reflect access to healthcare or the agreement to participate in the study based on ethnicity, rather than the prevalence of R-CPD in Black, White, Asian, and other populations. Currently, there is no epidemiological study investigating the ethnic background of patients with R-CPD symptoms.

### 3.2. The Potential Association with Comorbidities

Comorbidities were sometimes documented in studies, but a very low number of studies investigated potential associations. Chen et al. reported that 67.2% of R-CPD patients responding to the survey did not declare additional comorbidities [9]. Among others, 28.7% had gastroesophageal reflux disease (GERD); this proportion matches the GERD prevalence in the general Western population [17]. The association between GERD and R-CPD was additionally explored in nine studies (Table 2) [5,11,12,14,15,18,19,20,21].

Yousef et al. observed 38.5% GERD or laryngopharyngeal reflux disease (LPRD) in their study population [15]. Hiatal hernia was documented in 33% of R-CPD patients treated by Raymenants et al. [20], while Siddiqui et al. noted a history of GERD in 42.4% of patients, with 28.2% being treated with proton-pump inhibitor or H2 blocker therapy [18]. In the study of Mailly et al., 51.9% of patients had acid brash at the time of diagnosis [5]. In the same vein, Arnaert et al. found 50% of R-CPD patients with GERD, with 92.3% of patients reporting significant improvement of heartburn one month after botulinum toxin injection [11]. The high prevalence of GERD in R-CPD patients was, however, not corroborated by Anderson et al., who reported hiatal hernia and GERD in 34.7% and 16.3% of cases, respectively [19]. Regarding the significant differences between GERD and LPRD [23], Lechien et al. investigated LPRD rather than GERD in R-CPD patients and reported that R-CPD patients had a significantly higher Reflux Symptom Score-10 at baseline compared to individuals who could not burp without troublesome symptoms and controls [21]. Interestingly, the findings of this study did not support an increase in LPRD prevalence after botulinum toxin injection into the cricopharyngeal muscle, which remains a key protective mechanism against LPRD. Finally, in the pediatric population, Dorfman et al. documented GERD in 40% of R-CPD children, with 60% having proximal esophagus reflux events [12].

Despite high interest in understanding the role of reflux diseases in the development of R-CPD, there is currently no study investigating the occurrence of GERD and LPRD in childhood at the time of R-CPD onset. As suggested recently [24], retrograde cricopharyngeal hypertonicity may occur in children with severe GERD to protect the airway against reflux content. Future pediatric studies are needed to investigate R-CPD development in children with severe regurgitation or objectively diagnosed GERD and LPRD in childhood. Such studies should substantially improve the current R-CPD pathophysiological knowledge.

### 3.3. Functional Esophageal Findings

The esophageal physiology of R-CPD patients was investigated with high-resolution manometry (HRM) in six studies (Table 2) [12,14,15,19,20,22]. Studies commonly reported that R-CPD patients reported dysmotility disorders in 43.5% to 80.0% of cases, with absent peristalsis in 11% to 25% of cases. The mean basal upper esophageal sphincter (UES) and lower esophageal sphincter (LES) pressures ranged from 58.5 to 95.7 and from 6.3 to 20 mmHg, respectively. Heterogeneous results were found about the UES basal pressure differences across R-CPD and healthy controls (Table 2). HRM was used in two studies to confirm R-CPD diagnosis through carbonated drink provocative testing [20,22], with reports of consistent abnormalities, including esophageal pressurization abnormalities and failure of UES relaxation for retrograde gas. This test appears to be currently the most accurate approach for R-CPD diagnosis, incorporating the distal contractile integral as well [14,20]. Finally, the UES length was assessed by Yousef et al., who observed that R-CPD patients had longer cricopharyngeal sphincter length compared to controls (4.5 vs. 3.7 cm) [15].

In summary, HRM studies commonly suggest that traditional HRM may not identify R-CPD or abnormalities of UES tone when it is not considered through carbonated drink provocative testing. The moderate to high prevalence of non-specific dysmotility disorders may provide important information for differentiating asymptomatic R-CPD individuals vs. those with troublesome symptoms. Effective esophageal motility is commonly considered a defense mechanism against reflux disorder, including gastric content or air reflux (R-CPD). This may explain why some individuals with an inability to burp can effectively propel esophageal air into the lower digestive tract despite their inability to release it through the UES, thus remaining asymptomatic [24]. Because HRM is carried out during a small period of the day, it would be interesting to investigate the esophageal and sphincter motility in future studies with an experimental device capturing the occurrence of dysmotility events over a prolonged period.

### 3.4. Current Treatments Based on Botulinum Toxin and Future Insights

The treatment of R-CPD consists of the injection of botulinum toxin A into the posterior and lateral parts of the cricopharyngeal sphincter through an office-based or operating room approach. The operating room approach was initially described by Bastian and Smithson [3], but regarding the time and cost of this approach, many teams have performed office-based botulinum toxin injection (BTI) [4,16,21,25].

To date, three office-based BTI procedures have been described. The first is transcervical EMG-guided BTI [4,16,25], which has a reported effectiveness of 65 to 91% [4]. The second approach is transnasal BTI after a Valsalva maneuver to increase the retrocricoid space for the injection [26]. The third is the transtracheal approach in awake patients [27]. Both transnasal and transtracheal approaches are recent, and to date, there is no data on their effectiveness. Theoretically, the advantages of transnasal and transtracheal approaches include reduced risk of toxin spread to intrinsic laryngeal muscles and the ability to administer higher doses of botulinum toxin [26,27].

In terms of the overall effectiveness of BTI, a recent systematic review of 826 patients reported that the immediate success rate of BTI into the cricopharyngeal muscle in facilitating burping was 92.5% [4]. The effectiveness of BTI may depend on the approach used to perform the procedure; operating room BTI is associated with a much higher success rate than the office-based EMG-guided approach (Table 2) [4]. Importantly, the success/recurrence rate remains difficult to interpret across studies because of heterogeneity across studies in terms of dose and injection features. Insufficient dose in some cases/studies may result in early recurrence.

In skeletal muscle, the pharmacodynamic duration of botulinum toxin type A in tissues is approximately 3–6 months [28,29,30]. However, in tissues innervated by cholinergic autonomics (bladder and axillary glands), the duration is typically longer [31]. In R-CPD patients, a single BTI may lead to long-term symptom relief, which remains poorly understood. This finding was initially highlighted by Hoesli et al., who reported that 99% of patients experienced relief of the four cardinal symptoms (dose: 50–75 IU), with 79.9% experiencing relief of their symptoms beyond the pharmacologic duration of action after a single BTI into the cricopharyngeus muscle [8]. In the longitudinal study of Mailly et al., 75.5% of patients were treated with a single BTI (mean dose: 100 IU), while the authors identified family history of R-CPD as a negative predictor of single BTI success [5].

Recurrence occurred in the first month of follow-up in 9.5% of cases, whereas 12.6% and 27.9% of patients had recurrence during the 1–5 months and more than 6 months posttreatment, respectively. The single BTI effectiveness and the findings of this preliminary study may suggest that R-CPD patients can learn to burp, overcoming the retrograde UES dysfunction “reflex”. This hypothesis is indirectly supported by a recent paper demonstrating that speech therapy and a behavioral eructation retraining protocol may help patients to burp, especially when the medical treatment failed [32].

Because incomplete symptom relief may lead to a substantial relief of most symptoms in patients who continue to experience an inability to burp posttreatment, a valid and standardized patient-reported outcome measure is recommended to assess the pre- to post-BTI symptom frequency and severity [33,34,35]. To date, only the BURP score has been validated for assessing symptom changes throughout treatment (Figure 2) [36]. To the best of our knowledge, there is no additional study investigating the predictors of BTI effectiveness.

## 4. Conclusions

The pathophysiology of R-CPD remains primarily unknown despite significant quality-of-life impairments and an increasing number of studies investigating the esophageal physiology with HRM, impedance-pH testing, or barium swallow examinations. At first sight, the evidence suggests a potential genetic component (28% family history) and predominantly childhood onset (55.5%). The relationship between R-CPD and gastroesophageal reflux appears complex, with GERD prevalence ranging from 16.3 to 51.9% across studies. HRM has revealed dysmotility disorders in 43.5–80.0% of patients, suggesting R-CPD pathophysiology involves an interplay between cricopharyngeal dysfunction and esophageal motility. This review highlights several key pathophysiological insights while identifying substantial knowledge gaps.

## 5. Future Directions

Due to the recentness of R-CPD, a substantial number of findings are still unknown and require future prospective studies. First, genetic studies are needed to investigate and compare the genomes of R-CPD symptomatic patients from the same family, subjects unable to burp without symptoms, and healthy controls to identify the potential genetic background. Second, while the childhood and family history findings reported in numerous studies may suggest a potential genetic background, to date, without an epidemiological pediatric study, it remains difficult to exclude environmental etiologies. The potential link with reflux diseases highlights a potential interaction with the environment of patients; R-CPD members of the same family may be eating the same pro-reflux diet, leading to a family history of reflux that contributes to R-CPD, rather than a true genetic background. Further epidemiological studies could consider some children with pyloric stenosis and the related high proportion of regurgitation and reflux to investigate the post-disease occurrence of R-CPD and compare the R-CPD rate with a control group composed of healthy children. Third, the potential functional esophageal differences between mildly symptomatic subjects unable to burp and R-CPD patients need to be investigated using a prolonged study of esophageal motility. This point is particularly important given the short period assessed during a traditional HRM and the potential for missing dysmotility events throughout the day. Fourth, the effectiveness of a single BTI in most patients and the possibility of burping after a behavioral eructation retraining protocol need to be understood through large-cohort prospective studies. Clinical phenotype, dose, and technique need to be systematically studied to understand the effectiveness of potential therapy in future studies. 

## Figures and Tables

**Figure 1 toxins-18-00008-f001:**
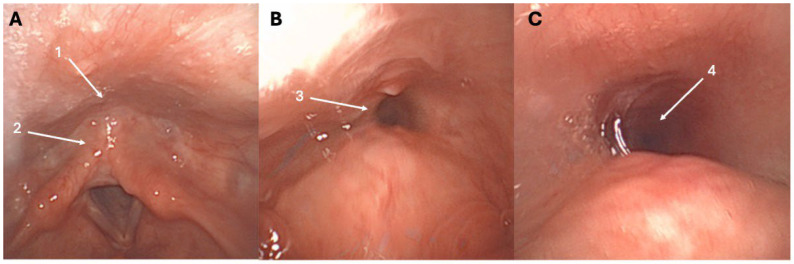
Tunnel sign throughout the nasofibroscopy. At rest, the entry of the esophagus (1) and retrocricoid region (2) appear normal (**A**). During swallowing, the tunnel sign may appear, reflecting the entrapped gas into the esophagus (3)—(**B**), (4)—(**C**), and could be a highly specific sign of R-CPD. The patient was a 27-year-old female with a childhood history of R-CPD, gurgling noises, chest pain, bloating, and flatulence.

**Figure 2 toxins-18-00008-f002:**
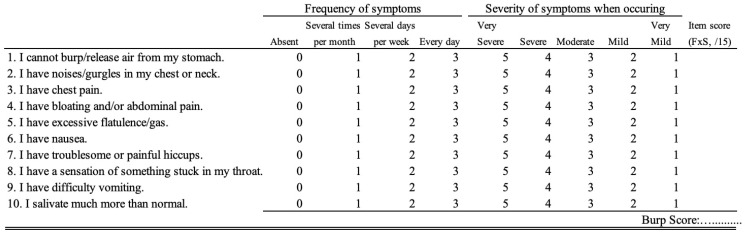
BURP score. An English version is presented. Currently, only the French version is validated [29]. The frequency and severity scores are multiplied to obtain a symptom score ranging from 0 to 15.

**Table 1 toxins-18-00008-t001:** Family and childhood features.

					Time of Occurrence		
References	Regions	Year	N	Family History	Childhood/Lifelong	Adolescence	Post-Adolescence
Arnaert [11]	Belgium	2024	50	NP	29 (58.0)	10 (20.0)	11 (22.0)
Mailly [5]	France	2024	106	18 (17.0)	28 (26.4)	0 (0)	0 (0)
Miller [10]	USA	2023	13	0 (0)	13 (100)	0 (0)	0 (0)
Chen [9]	International	2023	199	43 (21.6)	44 (22.7)	47 (24.2)	6 (3.1)
Dorfman [12]	USA	2023	5	NP	2 (40.0)	3 (60.0)	0 (0)
Hoffman [13]	USA	2022	5	NP	5 (100)	0 (0)	0 (0)
Oude Nijhuis [14]	Netherland	2021	8	NP	8 (100)	0 (0)	0 (0)
Hoesli [8]	USA	2020	200	84 (41.8)	196 (98)	4 (2)	0 (0)
			586	145/518 (28.0)	325/586 (55.5)	64/586 (10.9)	17/586 (2.9)

Outcomes consist of numbers and percentages/proportions. Abbreviations: NP = not provided; N = number.

**Table 2 toxins-18-00008-t002:** Functional outcomes.

Reference	Design	N	F/M	Age	Diagnosis	Functional Outcomes	Results
Sanagapalli	Prospective	52 R-CPD BTI	NA	NA	Abelchia	CDPT—esophageal pressurization patterns	R-CPD > CT *
2025 [22]	Controlled	7 R-CPD				CDPT—failure of UES relaxation	R-CPD > CT *
		6 CT				Posttreatment ability to burp	92% (3 months)
Anderson	Prospective	112 R-CPD	66/46	31.7	Abelchia	BS—Abnormalities	53%
2025 [19]	Uncontrolled					Hiatal hernia/GERD—Dysmotility–esophagitis	34.7–16.3–15.7%
						HRM abnormalities	67%
						IEM—absence of peristalsis	43.5%–17.5%
						UES/LES basal resting tone	85.3/15.2 mmHg
						Posttreatment ability to burp	NP
Yousef	Prospective	13 R-CPD	11/2	31.1	Abelchia	HRM findings	R-CPD vs. CT
2024 [15]	Controlled	26 CT	22/4	32.1		UES basal pressure	91.9 vs 49.7 mmHg *
						Ineffective swallow	70 vs. 15.4% *
						incomplete bolus clearance	81 vs. 21.8% *
						UES length	4.5 vs. 3.7 cm *
						GERD/LPRD treatment	5/13
Raymenants	Retrospective	55 R-CPD	30/25	28.5	Abelchia	Abnormal HRM	R-CPD-CT *-CT
2025 [20]	Case–Control	30 CT **	19/11	58.0	and one	IEM	54.5–30–33%
		15 CT	9/6	23.0	another	Absent peristaltism	11–17%—NP
					symptom	UES/LES basal resting tone (mmHg)	58.5/6.3 *–38.9/11.7 *–43/7 *
						Distal contractile integral (mmHg⋅cm⋅s)	146–577–316*
						Hiatal hernia	33–43–0%
						Belch provocation test	R-CPD > CT *
						Posttreatment ability to burp	NP
Dorfman	Prospective	5 Pediatric	5/0	16.4	Abelchia	UES relaxation (swallowing)	100% normal
2024 [12]	Uncontrolled	R-CPD				Proximal reflux event—dysmotility	60–80%
						GERD	40%
Oude Nijhuis	Prospective	8 R-CPD	4/4	27	Abelchia	UES pressure/gaseous reflux event	
2021 [14]	Uncontrolled				& other	IEM—absence of peristalsis	62.5–25%
					symptoms	Distal contractile integral (mmHg⋅cm⋅s)	237
					3 t/week	UES/LES basal pressures	95.7–20 mmHg
						Posttreatment ability to burp	100% (3 months)

* significant differences among groups; ** control patients who underwent HRiM for different conditions; Abbreviations: BS = barium swallow; CDPT = carbonated drink provocative testing; CT = control; F/M = female/male; GERD = gastroesophageal reflux disease; IEM = ineffective esophageal motility; LES = lower esophageal sphincter; N = number; NP = not provided; R-CPD = retrograde cricopharyngeus dysfunction; UES = upper esophageal sphincter; NA = not available.

## Data Availability

No new data were created or analyzed in this study.

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
