# Peer review of "Retrograde Cricopharyngeal Dysfunction: An Update of Pathophysiological Mechanisms and Future Directions"

_toxins, 2025, doi:10.3390/toxins18010008_

Round 1
Reviewer 1 Report
Comments and Suggestions for Authors
This is an important contribution on a condition for which characterization of the clinical phenotype and assessment of treatment options is emerging. The primary concern is the lack of cited data for some of the broader conclusions, as noted below. I understand that this is a difficult task, but I think that the authors should make another attempt.
Line 164: delete the "s" on "knowledges".
Line 211: the success rates did not appear clearly in Table 2, so the conclusion at the end of the sentence is difficult to cross-reference against the table.
Line 216: can you please define "lasting relief" - is there any way to quantitate this result in months or years?
Line 220: you will need to caveat that the success/recurrence rate is difficult to interpret in the absence of knowing doses and injection paradigm. In other words, "insufficient" dose would be expected to result in early recurrence. Nevertheless, until an effective dosing range can be determined, one can not easily assess the responder rate. If there are papers for which there is high quality data on dose and technique from which conclusions can be drawn, then please provide this in the results section.
Line 256: I am not sure that the word "discovery" applies. The disease is probably not "new", but has received recent attention, and only recently have investigators attempted systematic characterization.
Line 269: can you please define an "asymptomatic subject unable to burp"? If they are unable to burp, then are they not symptomatic?
Line 273: a major conclusion is that clinical phenotype, dose and technique need to be systematically studied to understand effectiveness of potential therapy.
Author Response
Reviewer 1:
This is an important contribution on a condition for which characterization of the clinical phenotype and assessment of treatment options is emerging. The primary concern is the lack of cited data for some of the broader conclusions, as noted below. I understand that this is a difficult task, but I think that the authors should make another attempt.
Line 164: delete the "s" on "knowledges".
Corrected.
Line 211: the success rates did not appear clearly in Table 2, so the conclusion at the end of the sentence is difficult to cross-reference against the table.
We have added, when available, the success rates of studies in Table 2 and the time of evaluation. See Table 2.
Line 216: can you please define "lasting relief" - is there any way to quantitate this result in months or years?
The duration of botox is 6 months. We have rephrased with the requested specification from the literature: “Notably, 75.5-79.9% of patients maintained symptom relief beyond the expected pharmacologic duration of botulinum toxin (approximately 6 months), suggesting potential neuroplastic adaptation or learned compensatory mechanisms in overcoming retrograde cricopharyngeal sphincter dysfunction.”
Line 220: you will need to caveat that the success/recurrence rate is difficult to interpret in the absence of knowing doses and injection paradigm. In other words, "insufficient" dose would be expected to result in early recurrence. Nevertheless, until an effective dosing range can be determined, one can not easily assess the responder rate. If there are papers for which there is high quality data on dose and technique from which conclusions can be drawn, then please provide this in the results section.
Agree. We have tempered the discussion and added the doses of the studies mentioning a clear success rate. Results, p.6, line 203: “The effectiveness of BTI may depend on the approach used to perform the procedure, the operating room BTI being associated with a much higher success rate than the office-based EMG-guided approach (Table 2) [4]. Importantly, the success/recurrence rate remains difficult to interpret across the literature studies because heterogeneity across studies in terms of dose and injection features. Insufficient dose in some cases/studies would be expected to result in early recurrence.
It is well-known that the duration of the botulinum toxin A in tissues is approximately 6 months [28-30]. In the R-CPD patients, a single BTI may lead to long-term symptom relief, which remains misunderstood. This finding was initially highlighted by Hoesli et al. who reported 99% of patients with relief of the four cardinal symptoms (dose: 50-75 IU) and 79.9% experienced relief of their symptoms beyond pharmacologic duration of action after a single BTI into the cricopharyngeus muscle [8]. In the longitudinal study of Mailly et al., 75.5% of patients were treated with a single BTI (mean dose: 100 IU), while authors identified the family history of R-CPD as a negative predictor of single-BTI success [5].”
Line 256: I am not sure that the word "discovery" applies. The disease is probably not "new", but has received recent attention, and only recently have investigators attempted systematic characterization.
We have modified: conclusion, p.10, line 1: “Due to the recentness of the R-CPD, a substantial number of findings are still unknown and require future prospective studies.”
Line 269: can you please define an "asymptomatic subject unable to burp"? If they are unable to burp, then are they not symptomatic?
We changed with mildly symptomatic.
Line 273: a major conclusion is that clinical phenotype, dose and technique need to be systematically studied to understand effectiveness of potential therapy.
The last line of the conclusion was modified to: “Clinical phenotype, dose and technique need to be systematically studied to understand effectiveness of potential therapy in future studies.”

Reviewer 2 Report
Comments and Suggestions for Authors
The manuscript is a scoping review about Retrograde Cricopharyngeal Dysfunction. The paper clearly and concisely summarises the current knowledge about this novel and rare clinical entity. I don't have any major issues to report. However, I have a couple of suggestions:
- The footnotes in Table 1 are empty. I suggest specifying that the number in brackets represents the percentage of cases.
- Lines 205–206, 'Avoidance of general anaesthesia': the EMG-guided transcervical approach does not require general anaesthesia, so 'avoidance of general anaesthesia' should only be considered an advantage with respect to the operating room approach.
Author Response
Reviewer 2:
The manuscript is a scoping review about Retrograde Cricopharyngeal Dysfunction. The paper clearly and concisely summarises the current knowledge about this novel and rare clinical entity. I don't have any major issues to report. However, I have a couple of suggestions:
- The footnotes in Table 1 are empty. I suggest specifying that the number in brackets represents the percentage of cases.
Corrected: Table 1 footnotes: “Outcomes consist of number and percentages/proportions. Abbreviations: NP=not provided; N=number.”
- Lines 205–206, 'Avoidance of general anaesthesia': the EMG-guided transcervical approach does not require general anaesthesia, so 'avoidance of general anaesthesia' should only be considered an advantage with respect to the operating room approach.
Corrected: line 194: “Theoretically, the advantages of transnasal and transtracheal approaches include reduced risk of toxin spread to intrinsic laryngeal muscles, and the ability to administer higher doses of botulinum toxin [26,27].”

Reviewer 3 Report
Comments and Suggestions for Authors
Retrograde cricopharyngeal dysfunction (R-CPD) is an unfamiliar disease. This paper describes its symptoms, epidemiology, pathophysiology, potential comorbidities, diagnostic examinations (including high-resolution manometry), and the efficacy of botulinum toxin injection (BTI) through a thorough literature review. The pathophysiology of R-CPD is especially informative for understanding this disease. Additionally, descriptions of BTI techniques and post-treatment outcomes are valuable for clinicians treating R-CPD.
The text is well-written. My only comment is that the footnote in Table 1 is missing.
Author Response
Reviewer 3:
Retrograde cricopharyngeal dysfunction (R-CPD) is an unfamiliar disease. This paper describes its symptoms, epidemiology, pathophysiology, potential comorbidities, diagnostic examinations (including high-resolution manometry), and the efficacy of botulinum toxin injection (BTI) through a thorough literature review. The pathophysiology of R-CPD is especially informative for understanding this disease. Additionally, descriptions of BTI techniques and post-treatment outcomes are valuable for clinicians treating R-CPD.
The text is well-written. My only comment is that the footnote in Table 1 is missing.
Thank you.
Corrected: Table 1 footnotes: “Outcomes consist of number and percentages/proportions. Abbreviations: NP=not provided; N=number.”

Reviewer 4 Report
Comments and Suggestions for Authors
This review, entitled "Retrograde Cricopharyngeal Dysfunction: An Update of Pathophysiological Mechanisms and Future Directions ", is very interesting and nicely presented.
The authors should clearly show the source of Figure 1: Was it taken from a patient by nasofibroscopy? details should be added about this patient and the procedure as a whole.
In the introduction section, authors should discuss the different aetiology and mechanisms of R-CPD.
In the discussion section, I suggest that the author state clearly the exact age at the occurrence of R-CPD instead of childhood/ adolescence
Author Response
Reviewer 4:
This review, entitled "Retrograde Cricopharyngeal Dysfunction: An Update of Pathophysiological Mechanisms and Future Directions ", is very interesting and nicely presented.
The authors should clearly show the source of Figure 1: Was it taken from a patient by nasofibroscopy? details should be added about this patient and the procedure as a whole.
Thank you. We have changed the title and footnotes: “Tunnel Sign Throughout the Nasofibroscopy.”
Figure 1 footnotes: At rest, the entry of esophagus (A1) and retrocricoid region (A2) appear normal (A). During swallowing, the tunnel sign may appear reflecting the entrapped gas into the esophagus (B3, C4) and could be a highly specific signs of R-CPD. The patient was a 27-year-old female with a childhood history of R-CPD, gurgling noises, chest pain, bloating and flatulence.
In the introduction section, authors should discuss the different aetiology and mechanisms of R-CPD.
The current literature is lacking about etiology. We have however changed the introduction accordingly: Introduction, p.2, line 49: “The tunnel sign reflects entrapped gas in the esophagus that cannot be eructated. To date, the etiology of R-CPD remains unelucidated, with recent studies suggesting a potential role of childhood reflux disease in cricopharyngeal and upper aerodigestive tract irritation, and related protective airway mechanisms (cricopharyngeal muscle hypercontractility) [4,5].”
In the discussion section, I suggest that the author state clearly the exact age at the occurrence of R-CPD instead of childhood/ adolescence
The exact age significantly varies across studies. Regarding the reviewer comment, we specified the median and global trend in the discussion, p.4, line 132: “Precisely, the exact age of first consultation is around 30 years, with many patients being diagnosed in their 20s and 30s.”

Round 2
Reviewer 1 Report
Comments and Suggestions for Authors
Thank you for replying to my comments and making the changes. I have just a couple comments:
Line 208: the exact duration of the BoNTA in neurons is not established in humans. I would change the sentence to: "In skeletal muscle, the pharmacodynamic duration of botulinum toxin type A in tissues is approximately 3-6 months."
Line 209: Please change "misunderstood" to "poorly understood". Then consider adding: "However, in tissues innervated by cholinergic autonomics (bladder, axillary glands), the duration is typically longer." This information is captured in the publications on axillary hyperhidrosis and overactive bladder - in the Phase 3 clinical trials of onabotulinumtoxinA, approximately 25% of patients experienced benefit greater than 12 months. You would need to cite the appropriate references.
Line 214: consider creating a new paragraph.
Author Response
Toxins
Brussels, December 2025.
Dear Professor,
I’m sending a revised manuscript dedicated to Laryngopharyngeal Reflux Disease entitled: “Retrograde Cricopharyngeal Dysfunction: An Update of Pathophysiological Mechanisms and Future Directions." which is submitted for publication in Toxins.
We thank the Editor for the suggestion. We considered all of them.
Thank you for replying to my comments and making the changes. I have just a couple comments:
Line 208: the exact duration of the BoNTA in neurons is not established in humans. I would change the sentence to: "In skeletal muscle, the pharmacodynamic duration of botulinum toxin type A in tissues is approximately 3-6 months."
Indeed. Done: “In skeletal muscle, the pharmacodynamic duration of botulinum toxin type A in tissues is approximately 3-6 months [28-30].”
Line 209: Please change "misunderstood" to "poorly understood".
Done: line 209= “In the R-CPD patients, a single BTI may lead to long-term symptom relief, which remains poorly understood”
Then consider adding: "However, in tissues innervated by cholinergic autonomics (bladder, axillary glands), the duration is typically longer." This information is captured in the publications on axillary hyperhidrosis and overactive bladder - in the Phase 3 clinical trials of onabotulinumtoxinA, approximately 25% of patients experienced benefit greater than 12 months. You would need to cite the appropriate references.
Done: “In skeletal muscle, the pharmacodynamic duration of botulinum toxin type A in tissues is approximately 3-6 months [28-30]. However, in tissues innervated by cholinergic autonomics (bladder, axillary glands), the duration is typically longer [31]. In the R-CPD patients, a single BTI may lead to long-term symptom relief, which remains poorly understood. »
New reference:
Lee DG, Kim JE, Lee WS, Kim MB, Huh CH, Lee YW, Choi GS, Lee JB, Yu DS, Shin MK, Roh MR, Ahn HH, Kim WS, Lee JH, Park KY, Park J, Lee WJ, Park MY, Kang H. A Phase 3, Randomized, Multi-center Clinical Trial to Evaluate the Efficacy and Safety of Neu-BoNT/A in Treatment of Primary Axillary Hyperhidrosis. Aesthetic Plast Surg. 2022 Jun;46(3):1400-1406. doi: 10.1007/s00266-021-02715-4.
Line 214: consider creating a new paragraph.
Done.
Thank you.
